# Genetic Diversity Analysis and Prediction of Potential Suitable Areas for the Rare and Endangered Wild Plant *Henckelia longisepala*

**DOI:** 10.3390/plants13152093

**Published:** 2024-07-29

**Authors:** Renfen Zhao, Nian Huang, Zhiyan Zhang, Wei Luo, Jianying Xiang, Yuanjie Xu, Yizhi Wang

**Affiliations:** 1College of Forestry, Southwest Forestry University, Kunming 650224, China; 18287524412@163.com (R.Z.); 18214665830@163.com (N.H.); swfuzzy@163.com (Z.Z.); 15208755725@163.com (W.L.); jy_xiang@hotmail.com (J.X.); 2Yunnan Academy of Biodiversity, Southwest Forestry University, Kunming 650224, China; 3College of Soil and Water Conservation, Southwest Forestry University, Kunming 650224, China

**Keywords:** *Henckelia longisepala*, ddRAD-seq, genetic structure, genetic diversity, potentially suitable areas

## Abstract

*Henckelia longisepala* (H. W. Li) D. J. Middleton & Mich. Möller is a rare and endangered plant species found only in Southeastern Yunnan, China, and Northern Vietnam. It is listed as a threatened species in China and recognized as a plant species with extremely small populations (PSESP), while also having high ornamental value and utilization potential. This study used ddRAD-seq technology to quantify genetic diversity and structure for 32 samples from three extant populations of *H. longisepala*. The *H. longisepala* populations were found to have low levels of genetic diversity (Ho = 0.1216, He = 0.1302, Pi = 0.1731, F_IS_ = 0.1456), with greater genetic differentiation observed among populations (F_ST_ = 0.3225). As indicated by genetic structure and phylogenetic analyses, samples clustered into three distinct genetic groups that corresponded to geographically separate populations. MaxEnt modeling was used to identify suitable areas for *H. longisepala* across three time periods and two climate scenarios (SSP1-2.6, SSP5-8.5). High-suitability areas were identified in Southeastern Yunnan Province, Northern Vietnam, and Eastern Laos. Future *H. longisepala* distribution was predicted to remain centered in these areas, but with a decrease in the total amount of suitable habitat. The present study provides key data on *H. longisepala* genetic diversity, as well as a theoretical basis for the conservation, development, and utilization of its germplasm resources.

## 1. Introduction

Gesneriaceae plants are highly appreciated by both domestic and international flower enthusiasts for their diverse morphology, vibrant and colorful flowers, and unique indoor ornamental characteristics [1]. From an early stage, certain developed countries, such as the United States, Canada, and Germany, have engaged in the development and research of Gesneriaceae plants. In addition to widely popular genera like *Saintpaulia* (commonly known as African violets) and *Sinningia*, numerous other genera and species have been cultivated, making them highly sought after in the current European and American markets [2]. Beyond their significant ornamental value, Gesneriaceae also hold important medicinal properties, particularly in the southern provinces of China such as Guangxi, Yunnan, and Guizhou. These plants are traditionally used in Chinese medicine for treating bruises, coughs, various sores, and poisons, with notable efficacy. For instance, species like *Hemiboea cavaleriei* and *Hemiboea subcapitata* are employed in their respective regions to treat conditions such as scabies and burns [3]. *H. longisepala*, a member of the Gesneriaceae family, is a subshrub with a stem height ranging from 25 to 80 cm and a stem diameter of 4–6 mm, and a short vertical rhizome in its underground part. Stems simple, apically densely brownish appressed when pubescent, basally glabrescent. Leaves opposite, equal to subequal in a pair; petioles 1.5–5.5 cm long, covered with long appressed brownish hairs when young, glabrescent; leaf blade ovate to oblanceolate, base attenuate to broadly cuneate, margins repand–crenulate to nearly entire, apex acuminate, glabrous to sparsely ciliate, especially abaxial on veins and especially when young, adaxially dark green, abaxially glaucous-green or sometimes dark purple; secondary veins 8–17 pairs; midvein and secondary veins prominent on both surfaces [4]. It has a beautiful plant shape, large and bright flowers, and high ornamental value.

In the wild, *H. longisepala* is found in the southeastern part of Yunnan Province, China (specifically in Hekou County, Jinping County, and Lüchun County), and Northern Vietnam, where it grows in valleys and shrublands at altitudes ranging from 250 to 800 m. It thrives within banana forests or in shaded areas along ditches. Its habitat requirements are highly specific, demanding stringent environmental conditions. Without a protective canopy from taller vegetation layers, *H. longisepala* individuals may fail to survive. The uniqueness of its habitat, coupled with its limited range and the ongoing destruction of valley rainforests, pose significant threats to the survival of existing *H. longisepala* populations. The threatened species list for China’s higher plants [5] lists *H. longisepala* as an endangered species; hence, its germplasm resources urgently need protection. Currently, one of the better-protected populations of *H. longisepala* is located in the Huanglianshan National Nature Reserve in Yunnan Province; the other populations are heavily anthropogenically disturbed, with their other habitats fragmented, resulting in lower numbers and poorer growth. The genetic diversity of a species, which is the product of historical evolution, supports its long-term survival and ability to adapt and change [6]. It is well known that preserving the genetic diversity of endangered species can benefit their long-term survival and evolution in changing environments [7,8]. Therefore, it is necessary for researchers to characterize the genetic diversity and population structure of endangered plant species, as a crucial step for their conservation and management [9].

Molecular marker technology is an important means of evaluating the genetic diversity of germplasm resources [10], whether of different populations of organisms or of different individuals from the same population [11,12]. Double digest restriction-site associated sequencing (ddRAD-seq) is a molecular marker technology developed by researchers to solve the problems of a relatively complicated library construction, high sequence dispersion, and high operation requirements [13,14,15,16]. ddRAD-seq utilizes multiplexed libraries obtained through the enzymatic digestion of whole genomic DNA followed by binding to specific adapters (creating reduced-representation libraries); it enables the acquisition of more SNP loci for a competitive cost [17]. ddRAD-seq technology not only improves sequencing efficiency, but also reduces the cost of experimentation [18]. Recently, ddRAD-seq technology has been widely applied in phylogenetics [19], molecular breeding [20], and population genetics [21], among other fields. Various enzymes suitable for ddRAD library construction in angiosperms have been described, enabling the rapid development of SNP markers and genome typing for multiple angiosperm species [22]. In addition, 18 QTLs (quantitative trait loci) closely related to yield traits in *Asparagus officinalis* were successfully identified using ddRAD, providing a solid theoretical foundation for the genetic analysis of yield traits and molecular marker-assisted breeding in *Asparagus officinalis* [23]. Furthermore, a study of *Rhododendron meddianum* with ddRAD technology revealed a high level of genetic diversity and moderate genetic differentiation among populations [6]. More recently, various studies have demonstrated that the ddRAD-seq technique can accurately determine SNP genotypes [24]. However, ddRAD-seq has not yet been utilized to study *H. longisepala* genetic diversity and genetic structure; to date, it has also not been used for any plant species in the same family (Gesneriaceae).

Owing to ongoing habitat destruction, *H. longisepala* populations are small and narrowly distributed. Understanding climate change impacts on *H. longisepala* distribution could provide valuable information for the conservation and management of this rare and endangered species. Climatic conditions are one of the most important factors affecting the natural geographical distributions of plant species, and climate change may lead to shifts in species’ distributions and also significantly impact their diversity [25]. Species distribution models are widely used in biogeography, conservation biology, and ecology because they can accurately predict the suitable habitat range for a given species [26]. The five most commonly used species distribution models are BIOCLIM, CLIMEX, ENFA, GARP, and MaxEnt [27]. The MaxEnt model is constructed based on known distribution points and associated environmental data, and then used to predict suitable areas based on relevant environmental variables [28]. Compared with other species distribution models, MaxEnt is simple to operate and can effectively predict suitable areas for narrowly distributed species with small sample sizes [29].

In this study, ddRAD-seq technology was used to study the genetic diversity and structure of *H. longisepala*, with the goal of better understanding its evolutionary potential and providing key reference data for its conservation. Using bioclimatic data and field distribution data for *H. longisepala*, MaxEnt modeling and ArcGIS software were used to predict the potential *H. longisepala* distribution under different climate scenarios. Key climatic factors affecting its distribution were identified to provide a theoretical basis for the conservation of *H. longisepala*.

## 2. Materials and Methods

### 2.1. Plant Materials and Data Collection

A total of 32 individual samples of *H. longisepala* were collected in January 2024 from three wild populations in the Hekou, Jinping, and Lüchun counties of Honghe Prefecture, Yunnan Province (Table 1). Several individuals were sampled per population, and one or two healthy, young leaves were selected per individual, and the distance between sampling points of the same populations was 50 m (between populations, it was more than 70 km). After collection, the leaves were quickly dried and preserved with silica gel. Further information on the sampling sites, number of samples, and population codes is provided in Table 1.

Distribution data were obtained from the Chinese Virtual Herbarium (https://www.cvh.ac.cn/), the Global Biodiversity Information Facility (https://www.gbif.org/zh/), the published literature, local flora guides, and field investigations. A total of 13 distribution records were obtained for *H. longisepala* after removing repetitive and unclear datapoints. Climatic data (bio1-19) were downloaded from the WorldClim database (https://worldclim.org/data/index.html, accessed on 13 April 2024); the data included 19 climate variables with a resolution of 2.5′ (about 5 km geographical resolution) for the current period (1970–2000), the 2030s (2021–2040), and the 2050s (2041–2060). Climate data for both future periods were predicted using the climate system model BCCCSM2-MR of The Beijing Climate Center (BCC) in the Global Climate Modeling (GCM) data block, a reliable source of future climate distribution data. For example, correlations between the model predictions and observed values were as high as 0.86 [28]. For the future climate data, there were four greenhouse gas emission scenarios: SSP1-2.6 (low-concentration CO_2_ emission scenario), SSP2-4.5 (low- and medium-concentration CO_2_ emission scenario), SSP3-7.0 (medium- and high-concentration CO_2_ emission scenario), and SSP5-8.5 (high-concentration CO_2_ emission scenario). Two greenhouse gas emission scenarios (SSP1-2.6, SSP5-8.5) were utilized in this study.

### 2.2. Genetic Diversity Assessment

#### 2.2.1. DNA Extraction, Library Construction, and Sequencing

Total genomic DNA was extracted from silica-dried leaf tissues using a modified CTAB method [30]. The quality of the extracted DNA was assessed via 0.8% agarose gel electrophoresis, and DNA samples were quantified using an ultraviolet spectrophotometer [31], thereby ensuring compliance with subsequent library-building requirements. A combination of DpnII and MspI endonucleases was used for the double digestion of the quality-inspected DNA samples. Based on the selected endonuclease, the most suitable reaction conditions were determined, typically setting the digestion temperature at 37 °C for 5 h. After enzyme digestion, 5 µL of each sample was taken for gel electrophoresis to verify digestion. The DNA enzyme digestion was required to be complete, meaning no main band should be visible post-digestion. Instead, there should be clearly dispersed bands between 250 and 1000 bp. VAHTSTM DNA Clean Beads (Vazyme, Nanjing, China) were then used to remove over- and under-sized fragments after digestion. Subsequently, the sheared fragments were ligated to the P2 adapter. A PCR was performed to enrich the sequencing library template using the DNA fragments connected to the adapters. Finally, the PCR products were purified using VAHTSTM DNA Clean Beads. Final fragment selection and purification of the DNA library was performed using 2% agarose gel electrophoresis (220~450 bp). After passing quality control, the libraries were paired-end sequenced on the Illumina NovaSeq platform by Shanghai Personalbio Co. (Shanghai, China).

#### 2.2.2. Sequencing Data Quality Control and Filtering

After sequencing, low-quality reads were removed using Fastp (v0.20.0) [32] to obtain high-quality sequencing data. The filtering criteria included the following: (1) The removal of adapter sequences, specifically at the 3′ end. (2) Quality filtering, using a sliding window approach, with a window size of 5 bp. Windows moved from the 3′ end to the 5′ end, and the average Q-value was calculated for each position; bases in the window were deleted if the Q-value was less than 20. (3) Length filtering: reads ≤50 bp in length were removed. (4) Fuzzy base N-filtering: reads with more than five N bases were removed.

#### 2.2.3. SNP Detection and Screening

As there is no published genome for the Gesneriaceae plant family, no reference genome was available for this study. Therefore, reads in each sample were clustered according to sequence similarity using ustacks (http://catchenlab.life.illinois.edu/stacks/, accessed on 26 April 2024) [33], and loci were combined across all samples using cstacks to obtain consensus sequences for each loci. SNPs were obtained using the populations program in Stacks. Finally, filtering in PLINK (v1.90b6) [34] was implemented to obtain high-quality SNPs (parameters: --geno 0.05 --mind 0.1 --maf 0.05).

#### 2.2.4. Data Analysis

The SNP data were analyzed with the following software. GCTA (v1.94.1) [35] was used to perform a principal component analysis based on the dataset of high-quality SNPs. The populations module of Stacks (v2.58) [33] was utilized to calculate genetic diversity parameters, including the expected heterozygosity (He), observed heterozygosity (Ho), the inbreeding coefficient (Fis), and nucleotide diversity (Pi). Admixture (v1.3.0) [36] was used to assess the population genetic structure with values of K ranging from two to ten (i.e., assuming the existence of 2–10 ancestral populations). A mixed model approach was used in Admixture, and the CV error was obtained for each value of K. The maximum likelihood algorithm in FastTree (2.1.11) [37] was used to construct a phylogeny for the *H. longisepala* samples (FastTree -gtr -nt alignment file --boot 1000 > tree file).

### 2.3. Suitable Area Prediction for H. longisepala

#### 2.3.1. Data Preprocessing

Distribution data for *H. longisepala* were processed in preparation for subsequent analyses. The longitude and latitude values for the distribution sites were converted into decimal degree values, input into Excel tables, and saved in ASC format for later use. 

After downloading the 19 climatic factors from Worldclim (Table 2), the prediction area was set to include Yunnan and the surrounding areas of Guangxi, Vietnam, and Laos. Climate data for the prediction area were obtained via mask extraction and saved in ASC format. 

#### 2.3.2. Correlation Analysis of Climatic Factors

As many climate variables are strongly correlated, their joint inclusion in statistical models can affect model outcomes, such as assessments of variable contributions and observed relationships among variables, thereby affecting the accuracy of the model fitting [38]. The bioclimatic data for each period included 19 climatic factors related to temperature and precipitation (Table 2). Therefore, an initial model was run in MaxEnt to evaluate the contributions of each climatic factor to the model predictions. Climatic factors with a contribution rate < 0% were removed in order of their rank. Next, a correlation analysis was performed in ENMTools (https://rdocumentation.org/packages/ENMTools/, accessed on 2 May 2024) for the remaining climatic factors. Highly correlated (|r| > 0.8) factors were grouped into pairs, and the variable with the lowest contribution rate was eliminated from each pair.

#### 2.3.3. Parameter Selection

The *H. longisepala* climate and distribution data were imported into MaxEnt, where response curves were drawn for each climatic factor, predicted distribution maps were created, and a variable importance analysis was conducted using the jackknife method. The model was run ten times, with 25% of the data points included in the validation set. The accuracy of the model predictions was verified by comparing the actual distribution points in the validation set with the predicted values. The receiver operating characteristic (ROC) curve generated by the MaxEnt (https://biodiversityinformatics.amnh.org/open_source/maxent/, accessed on 1 May 2024) model was used to evaluate the model accuracy, with the area under the curve (AUC) being used to measure the impact of the climatic variables on *H. longisepala* distribution predictions. The larger the AUC value, the greater the influence of a given variable, indicating the greater accuracy of the prediction results. The AUC ranges from zero to one, with AUC values from 0.5 to 0.6 indicating a failed prediction, 0.6 to 0.7 indicating poor prediction results, 0.7 to 0.9 good prediction results, and AUC values greater than 0.9 indicating a high level of confidence in the predictions [39].

#### 2.3.4. Classification and Description of Suitable Areas

The ArcGIS (https://www.esri.com/en-us/arcgis/products/arcgis-desktop/resources/, versions 10.7, accessed on 26 May 2024) reclassification function was used to assess suitability for *H. longisepala* based on the MaxEnt simulation results. Four suitability classes were established based on probability values, and the range of suitability values was determined via the equal distance method [40]. Non-suitable areas ranged from 0 to 0.2142, low-suitability areas from 0.2142 to 0.4284, moderate-suitability areas from 0.4284 to 0.6426, and high-suitability areas from 0.6426 to 1. The proportion of each study site suitable for *H. longisepala* was determined using a regional statistical function.

## 3. Results

### 3.1. ddRAD Sequencing and Data Processing

Double digest restriction-site associated (ddRAD) sequencing was performed for 32 *H. longisepala* individuals, and a total of 607,138,062 high-quality reads (or 18,973,064 per sample) were obtained after quality control (Table 3). High-quality reads accounted for more than 96% of all reads per sample, with an average of 97.96%. The average GC content per sample was 40.33%, and the average Q30 was 97.29%. The low GC content and high Q30 suggest that the library construction was successful, and the amount of data obtained met the requirements for subsequent data analyses.

### 3.2. SNP Statistics

SNPs were screened and filtered using PLINK [31], and a total of 1,613,563 SNPs were obtained, with an average of 50,423 valid SNPs per sample. More SNPs were heterozygous than homozygous: there were 822,245 heterozygous SNPs (or 52.87% heterozygous sites on average) and 791,318 homozygous SNPs (47.13%). Transitions or transversions occurred at 1,613,563 SNPs, with transitions being more common than transversions. The ratio of transitions to transversions (Ts/Tv) ranged from 1.31 to 1.48, with an average of 1.37. There were 934,743 SNPs with purine-to-purine or pyrimidine-to-pyrimidine transitions, with a transition ratio of 57.93%, and 678,820 SNPs with purine-to-pyrimidine substitutions, with a transversion ratio of 42.07% (Table 4).

### 3.3. Genetic Diversity Analysis

Heterozygosity is an important index of genetic variation in natural populations, which can reveal the population structure and even historical dynamics [41]. The observed heterozygosity (Ho) of the three *H. longisepala* populations ranged from 0.1182 to 0.1256 (mean Ho = 0.1216), and the expected heterozygosity (He) ranged from 0.1151 to 0.1883 (mean He = 0.1302) (Table 5). In the HSL and TMC populations, the observed heterozygosity was lower than the expected heterozygosity, indicating a history of self-fertilization in these populations. In the NXC population, the observed heterozygosity was greater than the expected heterozygosity, suggesting a heterozygote advantage. The inbreeding coefficient (F_IS_) ranged from 0.0420 to 0.207, with an average value of 0.1456 (F_IS_ > 0.78%); F_IS_ was highest in HLS, followed by TMC and NXC. This suggests a high level of inbreeding in some populations. Nucleotide diversity (Pi) is a measure of within-population genetic diversity and reflects the average variation between samples [42]. The nucleotide diversity of the three populations ranged from 0.1408 to 0.1990, with an average value of 0.1731. Among populations, Pi was highest in HLS; in other words, differences among HLS samples were more pronounced than for samples from other populations.

The population differentiation index (F_ST_) quantifies changes in population structure, and the magnitude of F_ST_ is correlated with the level of genetic differentiation, which may be affected by mutation, drift, and natural selection, among other factors [43,44]. The F_ST_ was calculated for each population pair (Table 6), with values ranging from 0.1924 to 0.4071 and a mean of 0.3225. While the HLS and TMC populations were well-differentiated (F_ST_ of 0.1924), genetic differentiation was highest between NXC and the other two populations (FST values greater than 0.25). The genetic distance was calculated between populations of *H. longisepala* (Table 7), with values ranging from 0.4524 to 0.8276 and a mean value of 0.6824. While the genetic distance between the HLS and TMC populations was the highest (0.8276), the genetic distance between the HLS and TMC population was the smallest (0.4524). The genetic distance between different samples ranged from 0.0072 to 0.1869 (S1), with an average value of 0.0795. The genetic distance between the TM-B6 and TM-C2 samples was the smallest, while the genetic distance between the NX-3 and TM-C6 samples was the highest (Appendix A).

### 3.4. Genetic Structure Analysis

Phylogenetic trees can be used to show the evolutionary relationships among populations within a species and to measure how closely related the populations are [45]. As can be seen from the phylogenetic tree for *H. longisepala* (Figure 1), the 32 samples were classified into three clusters, one for each population, and there was no evidence of intermingling between populations. 

In population genetic studies, SNP data can be used to classify individuals into subgroups via principal component analysis [42]. In this study, a principal component analysis (PCA) was performed on all 32 samples (Figure 2), with the first principal component (PC1) and second principal component (PC2) accounting for 27% and 16.7%, respectively, of the variation. Together, PC1 and PC2 categorized the samples into three groups, one for each population, which were widely spaced in the PCA diagram and genetically distinct.

Characterizing the population genetic structure can reveal ongoing evolutionary processes within a species [46]. In population structure analyses, CV error values are compared for different values of K; the smaller the CV error, the more accurate the K value. As shown in Figure 3, the CV error was minimized for K = 3, indicating that the optimal number of clusters for the study samples is three. When K = 2, NXC and HLS were grouped together, separately from TMC, which formed its own group. When K = 3, the three populations each represented their own group (Figure 4), although there was evidence of gene exchange between some HLS and TMC samples. The genetic structure results are consistent with those of the phylogenetic tree and principal component analysis.

### 3.5. MaxEnt Model Prediction and Major Climate Factors

Using MaxEnt, AUC values were obtained to monitor the simulation progress over ten repetitions. Under current climate conditions, the training set AUC for the potential distribution model was 0.974. Under the SSP1-2.6 scenario, the average AUC of the training set was 0.922 for the 2030s and 0.903 for the 2050s. Under the SSP5-8.5 scenario, the average AUC of the training set measured 0.925 in the 2030s and 0.919 in the 2050s. According to standard AUC evaluation criteria, the prediction model therefore achieved a high level of accuracy, effectively predicting the *H. longisepala* potential distribution with accurate and reliable simulation results.

Through the initial modeling and correlation analysis, this study ultimately identified seven bioclimatic factors suitable for constructing a MaxEnt model (Table 8). Among these seven factors, bio18 (precipitation of warmest quarter), bio7 (annual temperature range), bio16 (precipitation of wettest quarter), and bio9 (mean temperature of driest quarter) made significant contributions, with a cumulative contribution rate of 91.5%. Precipitation in the warmest quarter (bio18) had the greatest impact on the plant, with a contribution of 36.7%.

### 3.6. Current Potential Distribution of H. longisepala

Species distribution data generated by the MaxEnt model were imported into ArcGIS to create a geographical distribution map of suitable areas for *H. longisepala* under current climatic conditions (Figure 5). Predicted values were classified into four habitat suitability groups: non-suitable, low suitability, moderate suitability, and high suitability. The amount of habitat encompassed by each group was calculated. The potential *H. longisepala* habitat covered about 32.42 × 10^4^ km^2^, primarily in Southeastern Yunnan Province, Western Guangxi Province, Northern Vietnam, and Eastern Laos. Within this area, low-suitability habitat covered 15.67 × 10^4^ km^2^ (or 48.33% of the total suitable area), moderate-suitability habitat covered 11.25 × 10^4^ km^2^, and high-suitability habitat covered 5.50 × 10^4^ km^2^.

### 3.7. Potential Future Distribution of H. longisepala

Climate values for the 2030s and 2050s, under either the SSP1-2.6 or SSP5-8.5 scenario, were used to predict the future distribution of *H. longisepala* (Figure 6). Under the SSP1-2.6 scenario, the total suitable area covered 26.63 × 10^4^ km^2^ from 2021 to 2040, which is 17.86% less than the current habitat size (Table 9). Low-suitability habitat covered 14.12 × 10^4^ km^2^, moderate-suitability habitat 7.26 × 10^4^ km^2^, and high-suitability habitat 5.24 × 10^4^ km^2^; these values represent decreases of 9.89%, 35.47%, and 4.73%, respectively, from current values. From 2041 to 2060, suitable *H. longisepala* habitat covered 27.79 × 10^4^ km^2^, a decrease of 14.28% compared with the current value. Low-suitability habitat covered 14.93 × 10^4^ km^2^ and moderate-suitability habitat covered 7.76 × 10^4^ km^2^, a decrease of 31.02%. High-suitability habitat covered 5.10 × 10^4^ km^2^, a decrease of 7.27%. 

Under the SSP5-8.5 scenario, the total *H. longisepala* habitat covered 19.78 × 10^4^ km^2^ from 2021 to 2040, which is 38.99% less than the current habitat size. Low-suitability habitat covered 10.76 × 10^4^ km^2^, moderate-suitability habitat 4.99 × 10^4^ km^2^, high-suitability habitat 4.03 × 10^4^ km^2^; these represent reductions of 31.33%, 55.64%, and 26.73%, respectively, from current values. From 2041 to 2060, the total habitat area was 25.27 × 10^4^ km^2^, which is 22.05% less than the current habitat size. Low-suitability habitat covered 12.63 × 10^4^ km^2^, moderate-suitability habitat 7.66 × 10^4^ km^2^, and high-suitability habitat 4.98 × 10^4^ km^2^, representing decreases of 19.40%, 31.91%, and 9.45%, respectively, from current values. For both future emission scenarios, the total amount of suitable habitat for *H. longisepala* decreased, with suitable areas primarily located in Southeastern Yunnan Province, Northern and Western Vietnam, Eastern Laos, and Western Guangxi Province. The amount of high-suitability habitat also decreased, indicating that the future climate is predicted to have a negative impact on *H. longisepala* growth.

## 4. Discussion

### 4.1. Genetic Diversity

Breeding and improvement work depend upon adequate germplasm resources, but the effective utilization of germplasm resources relies on a characterization of their genetic diversity and other background studies [47]. In this study, genetic variation within and among populations of *H. longisepala* was investigated for the first time using ddRAD-seq. The genetic diversity within a species is influenced by a number of factors, including its breeding system, evolutionary history, geographic range, life history, and mode of seed dispersal [7]. Within populations, the observed heterozygosity averaged 0.1216, which was lower than the expected heterozygosity (0.1302), indicating the occurrence of self-fertilization within populations. Genetic diversity in *H. longisepala* (0.1731) was lower than that of *Ramonda myconi* [48], as assessed with RAPD (H = 0.2591, I = 0.3883). To date, only older molecular markers such as AFLPs and ISSRs have been utilized for the study of genetic diversity in the Gesneriaceae. For example, ISSRs were used to quantify genetic diversity in *Oreocharis benthamii* (H = 0.1014, I = 0.1528) [49], and AFLPs were utilized in *Primulina tabacum* (H = 0.220, I = 0.32) [50]. In general, the genetic diversity of endangered and very small populations of plants is low, as reported for *Vatica xishuangbannaensis* (0.169; RAPD markers) [51], *Bretschneidera sinensis* (0.141; ISSR markers) [52], and *Horsfieldia pandurifolia* (0.152; AFLPs) [53]. The low level of genetic diversity observed here for *H. longisepala* may be attributed to its highly specific habitat requirements, restricted pollen movement, and limited seed dispersal. According to the field investigation, the natural distribution of *H. longisepala* is narrow, with plants primarily located alongside highways, either under plantain forests or in the shade of a ditch. Seeds are oval, about 0.25 mm long, and frequently do not germinate after dispersal, such as when they fall into thick litter. Therefore, *H. longisepala* seedlings are few and mostly distributed along roads. These findings may also be due to the size of the study sample and its geographical attributes.

### 4.2. Genetic Structure

In this study, the principal component analysis, population genetic structure analysis, and phylogenetic analysis revealed that the three *H. longisepala* populations (HLS, TMC, NXC) were genetically separated concordant with their geographical distribution; little genetic exchange occurred among the sample populations during their evolutionary history. This pattern may be due to the limited number of individuals sampled and the small sampling range; for example, only four samples were collected in NXC. It may also be due to the narrow natural distribution of *H. longisepala* and obvious geographical isolation among populations. In addition, the FST among populations ranged from 0.1924 to 0.4071, with a mean value of 0.3225, indicating a high level of genetic differentiation among populations. The geographical distance between the study populations was more than 50 km, making it difficult for gene flow (as seeds or pollen) to occur between populations. Furthermore, according to the field surveys, the *H. longisepala* habitat was found to be severely disturbed by human activities. When the gene flow is low, it can produce genetic differentiation among populations [54]. The analyses in this study were complementary, and together indicated that the observed population structure was a reliable finding.

### 4.3. Prediction of Potential Suitable Areas

The geographic distribution of a given species is determined by a variety of factors, including anthropological, biological, and environmental factors, as well as the species’ own ability to evolve, spread, and adapt [55]. Climate is considered to be the most important factor shaping species’ distributions [56,57]. As documented here, the *H. longisepala* distribution is concentrated in the Southeastern Yunnan Province and Northern Vietnam. The predicted distribution indicated a wider habitat range for *H. longisepala* than that documented, and in addition to the existing records, there are potentially suitable areas for this species in Western Guangxi, as well as in Laos. According to the modeling results, precipitation in the warmest quarter, annual temperature range, precipitation in the wettest quarter, and the mean temperature of the driest quarter were the most important factors shaping *H. longisepala* distribution, indicating that rainfall and temperature were limiting factors in determining habitat suitability. *H. longisepala* distribution falls within a subtropical monsoon climate with high rainfall and temperatures, consistent with the observed *H. longisepala* distribution described here.

The prediction accuracy of the MaxEnt modeling depends on both the actual distribution data and environmental factors [58]. As a rare and endangered species, there are few reliably documented *H. longisepala* sites in the wild. Human activity also substantial affects *H. longisepala* survival, but no metric of human disturbance was included in this study, which may have affected the accuracy of the simulation results. At present, *H. longisepala* only exists in fragmented habitat patches in the wild, the availability of suitable habitat is continuously declining, and there is no current management scheme to introduce seedlings to new areas. Therefore, using study predictions, *H. longisepala* could be introduced to high-suitability areas as part of a conservation scheme. Future studies should examine aspects of *H. longisepala* physiology and biochemistry, tissue culture techniques, and relevant ecological characteristics.

### 4.4. Conservation Implications

The ultimate goal of species’ conservation is to ensure the continuous survival of populations and to maintain their evolutionary potential by preserving natural levels of genetic diversity [59,60]. The conservation status of *H. longisepala* in China has been determined in the present study by understanding the population structure and genetic variation between and within different populations. An in situ conservation plan should be implemented, with a prime focus on protecting the core areas with vulnerable populations, to preserve and maintain the prevailing genetic diversity. The ex situ approach of conservation should involve the establishment of a germplasm or seed bank for *H. longisepala*. However, priority should be given to the populations harboring higher genetic diversity for seed bank development. Fortunately, the HLS population is located inside a national nature reserve, which is less disturbed by anthropogenic activities, has better growth conditions, and has larger populations. From the field survey, the natural distribution of *H. longisepala* is narrow, and its habitat mostly consists of highly disturbed areas along highways. Hence, conservation of the *H. longisepala* habitat is key to preventing further losses of genetic variation. Conservation strategies should consider genetic aspects, in view of the need to reduce genetic drift and inbreeding [61]. Conservation measures should also aim to reduce anthropogenic impacts on *H. longisepala* survival, as well as on other Gesneriaceae species; species in this family are typically characterized by narrow distributions, small populations, and specialized growth conditions. Considering the low genetic diversity and significant genetic differentiation in *H. longisepala*, we suggest that in situ conservation should be established in all existing populations (especially the NXC population) to protect the natural habitats and individuals of *H. longisepala*. However, the genetic diversity and numbers of individuals of *H. longisepala* cannot be increased by short-term in situ conservation. We should implement artificial breeding, to transplant large numbers of seedlings to the natural populations to increase the size of the population. Then, introductions might be carried out in suitable areas, as identified here, with future studies of plant growth and the development needed to support population recovery. Apart from the conservation strategies mentioned above, the sensitization of local people about the alarming genetic situation of this important endangered *H. longisepala,* and involving local bodies in framing policies and programs for proper plant management, are also highly recommended.

## 5. Conclusions

In this study, the genetic diversity and genetic structure of 32 individual *H. longisepala* samples were investigated for the first time using ddRAD-seq technology. Low levels of genetic diversity were found in the *H. longisepala* populations, with the greatest variation among samples occurring in the HLS population; more genetic differentiation existed among populations than within populations. Using population structure analyses and PCA, samples were clustered into three distinct genetic groups. MaxEnt modeling was implemented to determine the potential *H. longisepala* distribution under two climate scenarios (SSP1-2.6 and SSP5-8.5) during the current period, 2021–2040, and 2041–2060. Bio18 (precipitation of warmest quarter), bio7 (annual temperature range), bio16 (precipitation of wettest quarter), and bio9 (mean temperature of driest quarter) were the main factors affecting *H. longisepala* distribution, with a cumulative contribution rate of 91.5%. Precipitation in the warmest quarter (bio18) had the greatest impact on the plant, with a contribution of 36.7%. At present, high-suitability areas for *H. longisepala* are mostly found in Southeastern Yunnan Province, Northern and Western Vietnam, Eastern Laos, and Western Guangxi Province. In the future, the potential *H. longisepala* distribution is predicted to remain centered in these areas, but the amount of suitable habitat will be reduced, especially for high-suitability habitat. 

## Figures and Tables

**Figure 1 plants-13-02093-f001:**
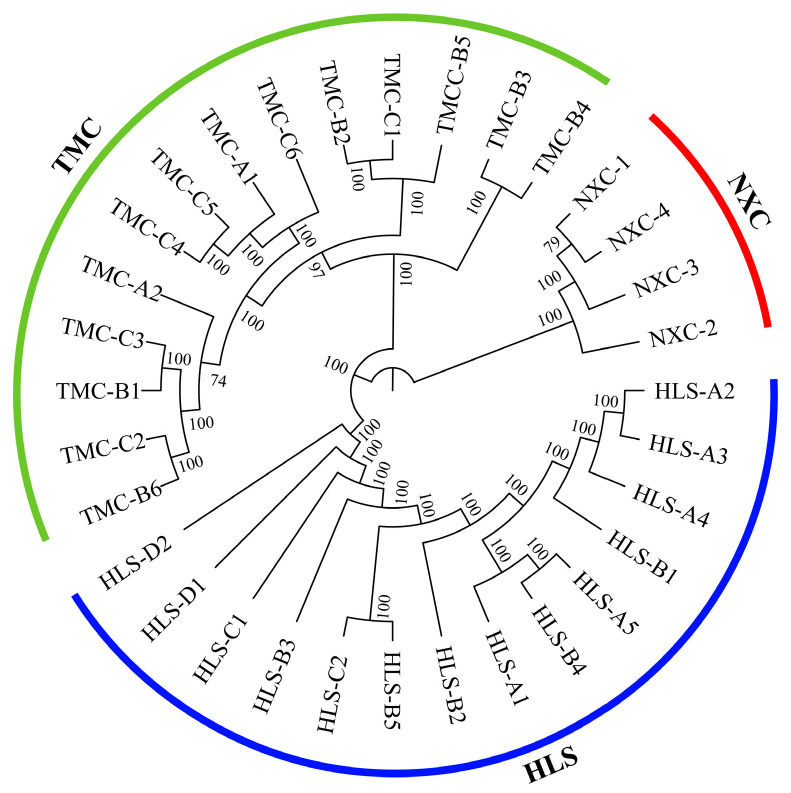
Phylogenetic tree for *Henckelia longisepala* samples based on SNP data, with 14 samples from the HLS population in blue; 4 samples from the NXC population in red; and 14 samples from the TMC population in green.

**Figure 2 plants-13-02093-f002:**
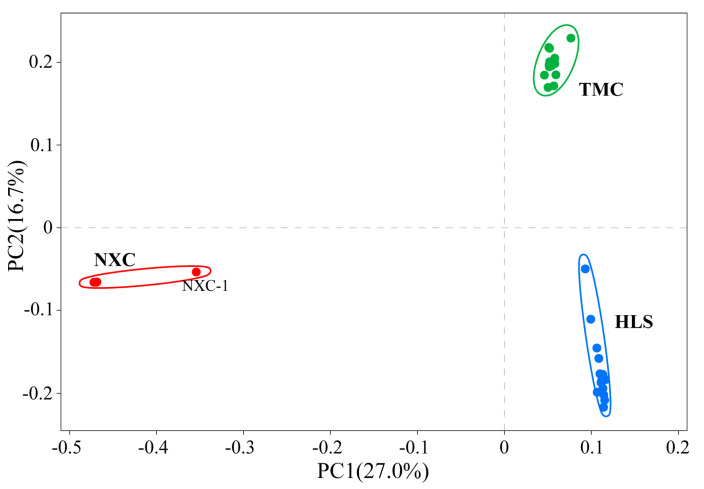
Principal components analysis (PCA) of all 32 *Henckelia longisepala* samples, with the proportion of the variance explained being 27% for PC1 and 16.7% for PC2.

**Figure 3 plants-13-02093-f003:**
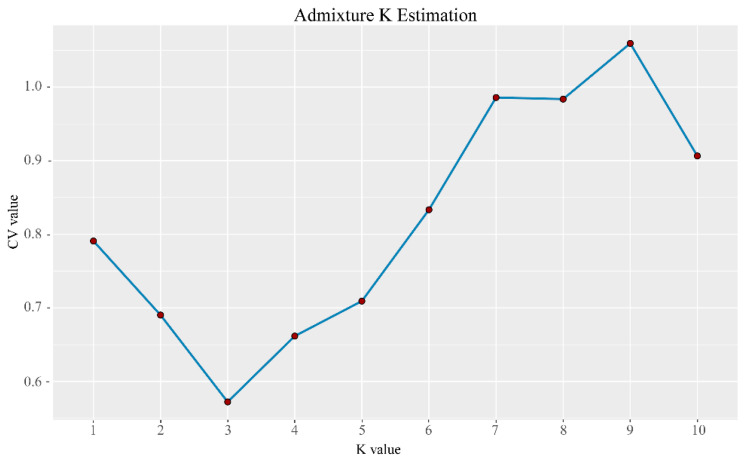
Distribution of CV error values corresponding to different K values; the CV error was minimized for K = 3, indicating that the optimal number of clusters for the study samples is three.

**Figure 4 plants-13-02093-f004:**
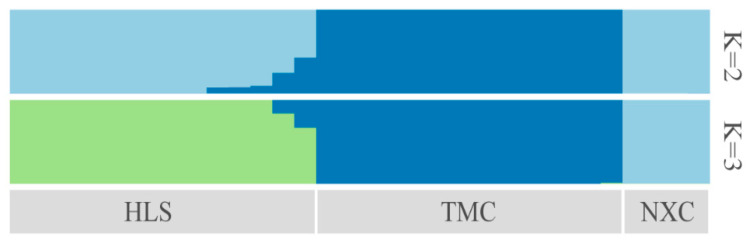
Genetic structure of *Henckelia longisepala* based on analysis; the results are K = 2 and K = 3.

**Figure 5 plants-13-02093-f005:**
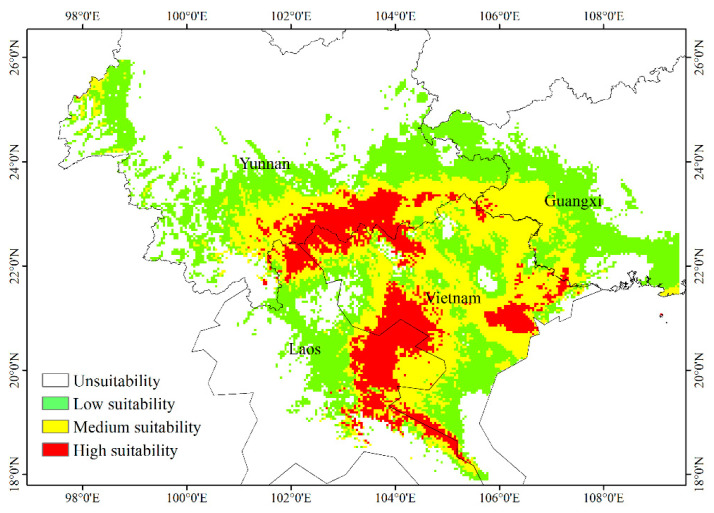
Potential geographic distribution of *Henckelia longisepala* in the current period.

**Figure 6 plants-13-02093-f006:**
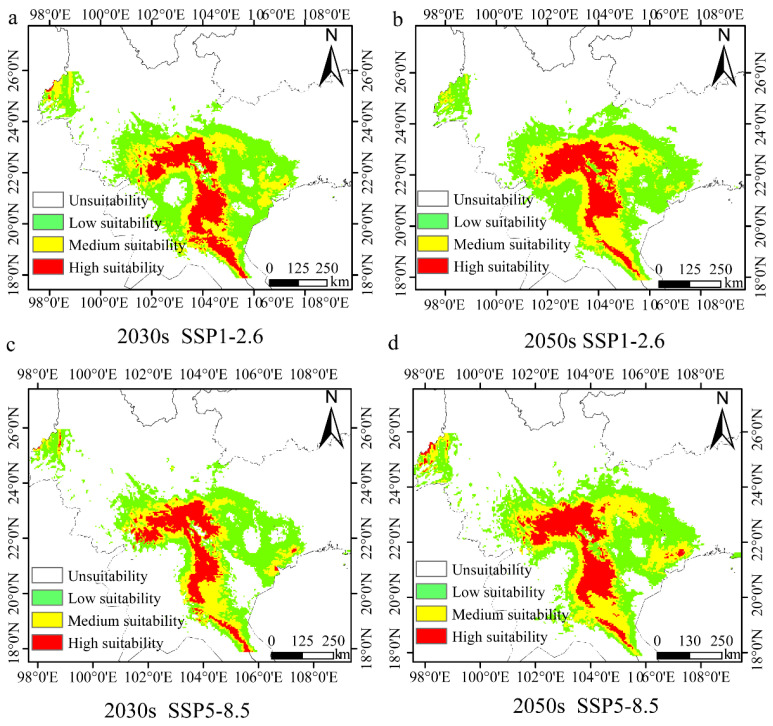
Potential geographic distribution of *Henckelia longisepala* for different time periods under two future climate change scenarios (SSP1-2.6, SSP5-8.5). Forecast maps are shown for the SSP1-2.6 climate change scenario in the 2030s (**a**) and 2050s (**b**) and for the SSP5-8.5 climate change scenario in the 2030s (**c**) and 2050s (**d**).

**Table 1 plants-13-02093-t001:** Collection site details for *Henckelia longisepala*.

Pop	Longitude (° E)	Latitude (° N)	Altitude (m)	Locality	Number of Samples
HLS	102.3494	22.6457	1009	Yunnan Lüchun	14
NXC	103.8897	22.6095	462	Yunnan Hekou	4
TMC	103.0264	22.6250	947	Yunnan Jinping	14

**Table 2 plants-13-02093-t002:** Overview of the 19 climatic variables.

Climate Variable	Description	Climate Variable	Description
bio1	Annual mean temperature	bio11	Mean temperature of coldest quarter
bio2	Mean diurnal range	bio12	Annual precipitation
bio3	Isothermality	bio13	Precipitation of wettest month
bio4	Temperature seasonality	bio14	Precipitation of driest month
bio5	Max temperature of warmest month	bio15	Precipitation seasonality
bio6	Min temperature of coldest month	bio16	Precipitation of wettest quarter
bio7	Temperature annual range	bio17	Precipitation of driest quarter
bio8	Mean temperature of wettest quarter	bio18	Precipitation of warmest quarter
bio9	Mean temperature of driest quarter	bio19	Precipitation of coldest quarter
bio10	Mean temperature of warmest quarter		

**Table 3 plants-13-02093-t003:** ddRAD sequencing data of *Henckelia longisepala*.

Sample_Name	HQ Reads	Total_Base (bp)	HQ Bases (bp)	HQ Bases %	GC (%)	Q30 (%)
HLS_A1	12,508,536	1,847,065,824	1,781,995,529	96.48	40.97	94.94
HLS_A2	14,332,744	2,120,032,512	2,041,843,119	96.31	43.12	94.92
HLS_A3	36,818,088	5,410,729,440	5,257,449,583	97.17	40.22	96.41
HLS_A4	34,107,414	5,009,117,184	4,872,148,968	97.27	39.26	96.19
HLS_A5	30,046,774	4,413,880,800	4,292,836,294	97.26	38.49	96.10
HLS_B1	32,220,506	4,733,449,056	4,602,590,836	97.24	39.03	96.29
HLS_B2	32,318,314	4,827,110,569	4,675,400,724	96.86	38.35	96.19
HLS_B3	32,792,914	4,906,151,916	4,745,839,024	96.73	38.04	96.00
HLS_B4	14,576,362	2,136,863,520	2,087,739,170	97.70	38.95	97.89
HLS_B5	15,292,504	2,244,107,808	2,189,839,488	97.58	39.19	97.82
HLS_C1	15,307,248	2,253,815,136	2,191,488,136	97.23	42.68	97.67
HLS_C2	18,084,322	2,656,833,696	2,589,175,360	97.45	42.18	97.71
HLS_D1	16,292,676	2,398,851,360	2,331,478,275	97.19	41.94	97.71
HLS_D2	16,406,160	2,414,157,408	2,347,565,196	97.24	43.57	97.66
NX_1	15,163,360	2,221,348,320	2,170,983,575	97.73	38.85	97.84
NX_2	17,780,208	2,614,287,456	2,545,830,403	97.38	39.43	97.73
NX_3	15,545,240	2,282,696,928	2,225,675,817	97.50	39.43	97.73
NX_4	14,758,408	2,167,298,496	2,112,253,626	97.46	38.15	97.72
TM_A1	14,494,752	2,130,803,712	2,076,078,685	97.43	43.20	97.82
TM_A2	16,837,792	2,481,160,320	2,410,517,148	97.15	43.16	97.59
TM_B1	14,951,876	2,195,641,152	2,140,724,678	97.50	39.91	97.74
TM_B2	14,226,138	2,093,088,960	2,037,690,222	97.35	42.23	97.68
TM_B3	14,467,912	2,133,347,904	2,071,285,801	97.09	41.71	97.65
TM_B4	14,766,900	2,175,391,872	2,115,346,007	97.24	41.65	97.70
TM_B5	13,956,478	2,053,735,200	1,999,096,791	97.34	40.38	97.66
TM_B6	15,690,434	2,297,819,808	2,247,479,371	97.81	39.11	97.97
TM_C1	15,504,368	2,272,298,112	2,218,737,176	97.64	38.23	97.86
TM_C2	17,817,304	2,612,644,992	2,550,693,042	97.63	39.69	97.88
TM_C3	16,814,138	2,466,775,008	2,407,381,170	97.59	40.25	97.84
TM_C4	17,449,764	2,558,971,008	2,495,422,887	97.52	41.10	97.80
TM_C5	18,474,068	2,713,303,584	2,644,282,676	97.46	39.40	97.75
TM_C6	17,334,360	2,539,707,264	2,482,303,852	97.74	38.78	97.87

**Table 4 plants-13-02093-t004:** SNP information for *Henckelia longisepala* samples from ddRAD sequencing.

Sample	SNP Number	Transitions	Transversions	Number of Heterozygous SNPs	Heterozygosity (%)	Number of Homozygous SNPs	Homozygosity (%)	Ts/Tv
HLS_A1	37,028	21,251	15,777	19,185	51.81	17,843	48.19	1.35
HLS_A2	37,915	21,857	16,058	21,569	56.89	16,346	43.11	1.36
HLS_A3	50,957	29,202	21,755	29,499	57.89	21,458	42.11	1.34
HLS_A4	49,246	28,245	21,001	28,974	58.84	20,272	41.16	1.34
HLS_A5	46,992	26,996	19,996	26,903	57.25	20,089	42.75	1.35
HLS_B1	48,142	27,603	20,539	27,442	57.00	20,700	43.00	1.34
HLS_B2	52,104	29,696	22,408	29,235	56.11	22,869	43.89	1.33
HLS_B3	42,699	24,250	18,449	23,372	54.74	19,327	45.26	1.31
HLS_B4	48,884	27,799	21,085	27,867	57.01	21,017	42.99	1.32
HLS_B5	48,644	27,821	20,823	23,862	49.05	24,782	50.95	1.34
HLS_C1	44,796	25,954	18,842	24,137	53.88	20,659	46.12	1.38
HLS_C2	50,281	28,952	21,329	29,745	59.16	20,536	40.84	1.36
HLS_D1	49,453	28,730	20,723	31,828	64.36	17,625	35.64	1.39
HLS_D2	43,263	25,069	18,194	20,529	47.45	22,734	52.55	1.38
NX_1	78,996	46,842	32,154	27,571	34.90	51,425	65.10	1.46
NX_2	65,403	39,064	26,339	17,770	27.17	47,633	72.83	1.48
NX_3	83,268	49,298	33,970	29,080	34.92	54,188	65.08	1.45
NX_4	79,208	46,794	32,414	27,658	34.92	51,550	65.08	1.44
TM_A1	40,827	23,647	17,180	22,512	55.14	18,315	44.86	1.38
TM_A2	42,953	24,843	18,110	24,407	56.82	18,546	43.18	1.37
TM_B1	37,417	21,613	15,804	22,603	60.41	14,814	39.59	1.37
TM_B2	40,854	23,637	17,217	22,332	54.66	18,522	45.34	1.37
TM_B3	40,333	23,211	17,122	23,883	59.21	16,450	40.79	1.36
TM_B4	43,117	24,863	18,254	25,114	58.25	18,003	41.75	1.36
TM_B5	38,974	22,396	16,578	21,086	54.10	17,888	45.90	1.35
TM_B6	47,327	27,388	19,939	27,724	58.58	19,603	41.42	1.37
TM_C1	40,341	23,226	17,115	18,166	45.03	22,175	54.97	1.36
TM_C2	48,286	27,923	20,363	29,165	60.40	19,121	39.60	1.37
TM_C3	48,601	27,859	20,742	30,902	63.58	17,699	36.42	1.34
TM_C4	43,142	24,796	18,346	24,096	55.85	19,046	44.15	1.35
TM_C5	48,577	27,991	20,586	29,012	59.72	19,565	40.28	1.36
TM_C6	95,535	55,927	39,608	35,017	36.65	60,518	63.35	1.41

**Table 5 plants-13-02093-t005:** Genetic diversity statistics for each *Henckelia longisepala* population.

Pop	Obs Het	Obs Hom	Exp Het	Exp Hom	Pi	F_IS_
HLS	0.1256	0.8744	0.1883	0.8117	0.1990	0.2079
TMC	0.1182	0.8818	0.1572	0.8428	0.1795	0.1868
NXC	0.1210	0.8790	0.1151	0.8849	0.1408	0.0420

**Table 6 plants-13-02093-t006:** Inter-population F_ST_ statistics.

Pop	HLS	TMC	NXC
HLS		0.1924	0.3680
TMC			0.4071

**Table 7 plants-13-02093-t007:** Genetic distance between populations of *Henckelia longisepala*.

Pop	HLS	TMC
TMC	0.4524	
NXC	0.8276	0.7672

**Table 8 plants-13-02093-t008:** Contributions of the bioclimatic variables to the MaxEnt model simulations.

Environment Variable	Percent Contribution	Permutation Importance
bio18	36.7	43.9
bio7	25.4	33.9
bio16	17.5	16.3
bio9	11.9	2.5
bio17	3.2	2.3
bio12	3.1	0.6
bio3	2.2	0.5

**Table 9 plants-13-02093-t009:** Amount of suitable *Henckelia longisepala* habitat predicted for different time periods (/10^4^ km^2^).

Scenario	Period	High Suitability	Ratio %	Medium Suitability	Ratio %	Low Suitability	Ratio %	Total	Ratio %
Current	—	5.50	—	11.25	—	15.67	—	32.42	—
ssp1-2.6	2030s	5.24	−4.73	7.26	−35.47	14.12	−9.89	26.63	−17.86
	2050s	5.10	−7.27	7.76	−31.02	14.93	−4.72	27.79	−14.28
ssp5-8.5	2030s	4.03	−26.73	4.99	−55.64	10.76	−31.33	19.78	−38.99
	2050s	4.98	−9.45	7.66	−31.91	12.63	−19.40	25.27	−22.05

## Data Availability

The original contributions presented in the study are included in the article and Appendix A; further inquiries can be directed to the corresponding author.

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
