# Peer review of "Genetic Diversity Analysis and Prediction of Potential Suitable Areas for the Rare and Endangered Wild Plant Henckelia longisepala"

_plants, 2024, doi:10.3390/plants13152093_

Round 1

Reviewer 1 Report

Comments and Suggestions for Authors

The MS is clear and well prepared alsthough it is not following instruction for the structure for Plants. 

However, I like the paper and I have minor suggestions:

Species latin names should contain also authorities where they appear firstly in the text. 

I suggest to make stronger part of Introduction on those what is known on species biology. This chapter is rather small in comparison with explaining molecular approach to Your study. Thus, al interesting data on species biology, with special emphasis on its survival, measures of conservation at present and programs that are in progress to species survival.

Also, I suggest to include chapter in concluding remarks where Your finding are affecting further conservation activities for this species. Some proposals and further investigations in this direction (e.g. which are the best places for introductions, reintroductions, which plant material, which genotypes, etc...

Author Response

1:. I suggest to make stronger part of Introduction on those what is known on species biology. This chapter is rather small in comparison with explaining molecular approach to Your study. Thus, al interesting data on species biology, with special emphasis on its survival, measures of conservation at present and programs that are in progress to species survival.

answer: many thanks for your comments. We have added some species biology in the Introduction. [please see lines33-40 and lines51-55 in the new MS]

2: I suggest to include chapter in concluding remarks where Your finding are affecting further conservation activities for this species. Some proposals and further investigations in this direction (e.g. which are the best places for introductions, reintroductions, which plant material, which genotypes, etc...

answer: many thanks for your comments. We have, accordingly, added some proposals and further investigations in this direction in the conservation. [please see lines445-454, lines441-450 and lines 457-469]

Reviewer 2 Report

Comments and Suggestions for Authors

Specify the ddRAD-seq protocol used.

What is the distance between sampling points of the same populations? What is the distance between populations? What do the affixes A-D and numbers in population names mean?

Provide a table of genetic distances between samples and populations.In the Results decrypt the most important climatic factors shaping the H. longisepala distribution.

What is an input for phylogenetic tree construction?. What are the parameters used for tree inferring? Was bootstrap analysis performed? If yes, indicate the support values on the tree. Expand legends for Figures 1-4. The figures should be fully understandable without reference to the text of the article.

Please provide links for Worldclim, MaxEnt, ENMTools, ArcGIS, and Ustacks.

Specify the manufacturer of VAHTSTM DNA Clean Beads.

Reviewer 3 Report

Comments and Suggestions for Authors

Why were only 4 samples taken from the NXC population?

151 The SNP data were analyzed with the following software. -
please list the software used

Author Response

  1. Why were only 4 samples taken from the NXC population?

Answer: The NXC population was more seriously disturbed by human activities, with fewer plants, and only four samples could be collected at distances of more than 50 m between plants.

  1. 2. 151 The SNP data were analyzed with the following software. - please list the software used

Answer: Thank you for pointing this out. we have modified accordingly in the new MS. The software used is as follows, Stacks (http://catchenlab.life.illinois.edu/stacks/)、ADMIXTURE (http://dalexander.github.i o/admixture/)、GCTA(https://yanglab.westlake.edu.cn/software/gcta/)、FasTree (https://bioinformaticsworkbook.org/phylogenetics/FastTree.html).